# Methylation Profile of Small Breast Cancer Tumors Evaluated by Modified MS–HRM

**DOI:** 10.3390/ijms241612660

**Published:** 2023-08-10

**Authors:** Aleksey M. Krasnyi, Alsu A. Sadekova, Vlada V. Kometova, Valeriy V. Rodionov, Ekaterina L. Yarotskaya, Gennadiy T. Sukhikh

**Affiliations:** 1National Medical Research Center for Obstetrics, Gynecology and Perinatology Named after Academician V.I. Kulakov of Ministry of Healthcare of Russian Federation, 117997 Moscow, Russia; 2Research Institute of Molecular and Cellular Medicine, Peoples’ Friendship University of Russia (RUDN University), 117198 Moscow, Russia; 3Department of Obstetrics, Gynecology, Perinatology and Reproductology, First Moscow State Medical University Named after I.M. Sechenov, 119991 Moscow, Russia

**Keywords:** breast cancer, small tumors, DNA methylation, *GCM2*, *ITPRIPL1*, *CACNA1E*, *DLGAP2*, biomarker, MS–HRM, pyrosequencing

## Abstract

The DNA methylation profile of breast cancer differs from that in healthy tissues and can be used as a diagnostic and prognostic biomarker. Aim of this study: To compare the levels of gene methylation in small malignant breast cancer tumors (<2 cm), in healthy tissue, and in fibroadenoma, and to evaluate the effectiveness of the modified Methylation Sensitive–High Resolution Melting (MS–HRM) method for this analysis. Analysis was performed using the modified MS–HRM method. For validation, the methylation levels of five genes were confirmed by pyrosequencing. The main study group included 96 breast cancer samples and the control group included 24 fibroadenoma samples and 24 healthy tissue samples obtained from patients with fibroadenoma. Breast cancer samples were divided into two subgroups (test set and validation set). The methylation of the following 15 genes was studied: *MAST1*, *PRDM14*, *ZNF177*, *DNM2*, *SSH1*, *AP2M1*, *CACNA1E*, *CPEB4*, *DLGAP2*, *CCDC181*, *GCM2*, *ITPRIPL1*, *POM121L2*, *KCNQ1*, and *TIMP3*. Significant differences in the validation set of samples were found for seven genes; the combination of the four genes *GCM2*, *ITPRIPL1*, *CACNA1E*, *DLGAP2* (AUC = 0.99) showed the highest diagnostic value based on logistic regression for all breast cancer samples. Our modified MS–HRM method demonstrated that small breast cancer tumors have a specific DNA methylation profile that distinguishes them from healthy tissues and benign proliferative lesions.

## 1. Introduction

In 2020, 2.3 million women were diagnosed with breast cancer, making it the most common cancer in the world [1]. It is known that in the early stages of cancer, including breast cancer, there are changes in the methylation of many genes [2].

DNA methylation occurs via the modification of DNA through the addition of a methyl group to the position 5′ of a cytosine that precedes a guanine (CpG). CpG is often found at a high density in the genome, forming CpG islands. The study of the methylation of CpG islands in promoter regions of genes is important because the hypermethylation of CpG islands can lead to the suppression of gene transcription—in particular, through the downregulation of tumor suppressor genes—and to the development of cancer. Aberrant DNA methylation in tumors as an early diagnostic and prognostic marker is undergoing vigorous research with high genome coverage using bioinformatic approaches, with the goal of its introduction into clinical practice.

The deficit of simple and easily reproducible methods for assessing the methylation of small DNA fragments limits the applications of methylation studies in routine clinical practice. A large number of barely reproducible methods for methylation studies have been suggested [3]. Some methods are periodically modified, as, for example, in the study by Aibel et al. [4]. The lack of unified methods assumes the absence of a generally accepted set of genes, which allows tumor tissue to be distinguished from healthy tissue. The most valuable results are those obtained in the studies with high coverage of the genome, primarily using BeadChip technology (Illumina, San Diego, CA, USA). The gene panels proposed by researchers show high accuracy, but the sets of genes in the various panels differ significantly; this may be due to small set of samples, which is typical for application of Illumina chips, or to the various bioinformatic approaches used and the lack of large validational sample set. This study focuses on gene methylation in small breast cancer tumors (<2 cm) compared to healthy tissue and fibroadenoma. We used our modified MS–HRM method, which makes it possible to evaluate the methylation of a large fragment of DNA (up to 700 base pair (bp)). This is important for determining the functional role of methylation. The methylation of large DNA fragments can cause gene suppression or silencing, while the methylation of a single site may cause significant functional effects. We have studied the methylation of genes that have been proposed as being of interest in high-coverage whole-genome sequencing studies. Only genes with CpG islands in promoter regions and/or the first exon were used.

We selected the genes *MAST1*, *PRDM14*, and *ZNF177* using a publication by Mao et al. [5] based on a bioinformatics analysis of publicly available datasets; the genes *DNM2*, *SSH1*, *AP2M1*, and *TIMP3* were selected from a similar article by Panagopoulou et al. [6]; the genes *GCM2*, *ITPRIPL1*, and *CCDC181* were selected from an investigation that included a bioinformatics analysis of publicly available datasets and the results of the research by Wang et al. [7]; the genes *CACNA1E*, *CPEB4*, and *DLGAP2* were selected from a publication by Luo et al. [8] based on a bioinformatics analysis of publicly available datasets followed by experimental confirmation of the results; the genes *POM121L2* and *KCNQ1* were selected from a paper studying methylation in normal healthy breast epithelium, which is a potential origin of breast cancer [9].

The aim of this study was to investigate the gene methylation profile in small (<2 cm) breast cancer tumors compared to healthy tissue and fibroadenoma. The results may broaden horizons for the development of diagnostic and prognostic test systems—the most promising of which might be a test system for the minimally invasive diagnosis of breast cancer based on blood plasma DNA examination.

## 2. Results

This study included 96 patients with breast cancer who were divided into two equal independent groups (test set and validation set). The control group included 24 patients with fibroadenoma. The age and BMI of the studied groups are presented in Table 1.

The tumor subtypes and histological characteristics of the breast cancer tumors of the test set and validation set are presented in Table 2.

The size of the fibroadenoma was 2 (1.5; 2.6) cm.

The methylation levels of the 15 genes *MAST1*, *PRDM14*, *ZNF177*, *DNM2*, *SSH1*, *AP2M1*, *CACNA1E*, *CPEB4*, *DLGAP2*, *CCDC181*, *GCM2*, *ITPRIPL1*, *POM121L2*, *KCNQ1*, and *TIMP3* were studied by modified MS–HRM in small malignant breast tumor (test set, n = 48), fibroadenoma (n = 24), and healthy tissue from patients with fibroadenoma (n = 24). No statistically significant differences in the methylation level of the studied genes were found between the fibroadenoma and healthy tissue samples. Therefore, these samples were included in the control group. Significant differences between the test set and the control group were found for eight genes: *CCDC181*, *GCM2*, *ITPRIPL1*, *ZNF177*, *CACNA1E*, *DLGAP2*, *TIMP3* (all *p* < 0.001), and *PRDM14* (*p* = 0.002; Figure 1).

The *DNM2*, *AP2M1*, *CPEB4*, and *SSH1* genes had only a non-methylated DNA fraction, which did not allow for the use of the MS–HRM method to determine the level of methylation. The results of the *POM121L2*, *KCNQ1*, and *MAST1* gene examination are presented in Table 3.

The validation set was used to assess methylation only in eight genes, showing significant differences for the test set. The validation set showed significant differences in seven genes: *CCDC181, GCM2*, *ITPRIPL1*, *ZNF177*, *CACNA1E*, *DLGAP2*, *PRDM14* (all *p* < 0.001), and *TIMP3 p* = 0.23 (Figure 1).

The diagnostic value of the genes in the test set and validation set, calculated according to the ROC analysis, is presented in Table 4.

We also combined the results of the test and validation sets and determined the optimal diagnostic gene set using logistic regression. This set included the *GCM2*, *ITPRIPL1*, *CACNA1E*, and *DLGAP2* genes. The AUC was 0.99 (0.97–1). To assess the model stability, we also calculated the CIs for the AUC using a bootstrap method [10]. This is a simple but powerful method for estimating confidence intervals without the need to repeat the experiment; it is achieved by constructing a number of different samples from the observed data set [11]. Using the bootstrap method, we obtained CI = 0.9675–0.9988, which confirms the stability of the AUC. The results are shown in Figure 2.

We used the Spearman’s rank correlation coefficient to compare the methylation levels of the studied genes in the general set and the characteristics of the tumors from which they were isolated. It was found that only *CCDC181* gene methylation was characteristic for LumA, while *PRDM14*, *CACNA1E*, *ITPRIPL1* gene methylation was specific for LumB. A negative correlation was also noted between the triple-negative subtype and the methylation of the *ITPRIPL1* and *CCDC181* genes. Only the relationship between the methylation level of the *TIMP3* gene and invasive apocrine breast cancer was significant. We found no relationship between lymph node metastasis and methylation. The level of methylation in four genes—*PRDM14*, *CACNA1E*, *CCDC181*, and *GCM2*—correlated with tumor size (Figure 3).

To confirm the validity of the modified MS–HRM method, we compared our results with the pyrosequencing results for five genes—*CCDC181*, *ZNF177*, *ITPRIPL1*, *GCM2*, and *DLGAP2*—using the Spearman’s rank correlation (r_s_). The amplicons of these genes were shortened by NEST-PCR to a length suitable for pyrosequencing methylation analysis. A significant correlation (*p* < 0.001) was found in all cases (Figure 4).

Pyrosequencing showed higher diagnostic values compared to modified MS–HRM (Table 5).

## 3. Discussion

Breast cancer, like other types of cancer, is associated with epigenetic changes. Assessments of epigenetic changes, such as DNA methylation, can provide important diagnostic and prognostic information. However, biomarkers of DNA methylation are not applied in routine medical practice. This can be explained by the absence of simple, generally accepted, and clinically applicable methods for the assessment of methylation that can be used in clinical practice. At the same time, an optimal combination of genes the methylation of which could provide us with important clinical data has not been defined yet.

A lot of data has been collected on the DNA methylation profile in breast cancer by means of chips such as the Infinium Methylation EPIC BeadChip (Illumina, San Diego, CA, USA) and through whole-genome methyl-sensitive sequencing. A number of authors have proposed breast cancer methylation profiles that they consider to be effective for tumor diagnosis.

In this study, we showed that our simple and cheap modified MS–HRM method can be used to validate results on a large set of samples obtained from whole-genome breast cancer methylation analyses. An important feature of the proposed method is the ability to evaluate long DNA fragments, which increases the diagnostic value of the MS–HRM. This is confirmed by a comparison of the AUC from Table 5 and Table 4. The results of the MS–HRM were confirmed by further pyrosequencing. However, pyrosequencing provided higher diagnostic accuracy than the MS–HRM; this can be explained by the presence of non-target products in the MS–HRM assay, while pyrosequencing is performed using NEST-PCR, which eliminates non-target products. When verifying the results obtained in the test set, we found that seven out of eight genes also showed high diagnostic significance, indicating their potential diagnostic value. Correlation analysis showed that the methylation of the *PRDM14*, *CACNA1E*, *CCDC181*, and *GCM2* genes correlated with tumor size. This may mean that during tumor growth, cells with methylated DNA for these genes increase in number faster than cells with unmethylated DNA. This suggests that these genes are associated with apoptosis or proliferation blocking. Indeed, methylation-mediated repression of *PRDM14* has been shown to promote apoptosis evasion [12], and Wang et al. noted that the hypermethylation of circulating *CCDC181* and *GCM2* is significantly associated with the overexpression of the proliferation marker Ki-67 [7]. It could be suggested that tumors with methylated DNA for the mentioned genes have more active proliferation typical of LumB−. The correlation analysis in our study showed that LumB- is characterized by the methylation of two genes—*PRDM14* and *CACNA1E*—which could be considered a potential prognostic marker of tumor progression.

It is important to note that the level of gene methylation in breast cancer depends on various factors, including tumor size and subtype. Thus, to develop a diagnostic model, it is necessary to take this dependence into account and use a larger set of validating samples to cover possible tumor variants. In our study, which included 96 cancer samples, we found that the AUC = 0.99 when assessing the methylation of four genes. To confirm the stability of the model, we calculated the CIs for the AUC using a bootstrap method. We showed that our model is stable and does not depend on the choice of samples.

The selection of patients for the control group is an important aspect of studies. In many studies, healthy tissue from patients with breast cancer is used as a control. However, the possibility of cancer cells migrating to adjacent healthy tissue should not be ignored. Our control group was comprised of patients with fibroadenoma; this allowed us to compare methylation in both healthy tissue and in a benign tumor.

A significant challenge for cancer management is neoplasm detection at an early stage; for this, the study of patients’ plasma may be helpful. Several clinical studies have been conducted so far—however, they have shown low sensitivity for early stages; for example, the sensitivity of the Multi-cancer early detection test in symptomatic patients referred for cancer investigation in England and Wales (SYMPLIFY) was found to be only 37.3% for stages I–II [13]. The study of the methylation of the genes proposed by us in plasma for the diagnosis of breast cancer may be effective when using modified MS–HRM, as it is known that methylated DNA fragments are longer than unmethylated ones [14]; the study of only long DNA fragments would increase the sensitivity of the method.

## 4. Materials and Methods

The study included 96 patients with breast cancer, who were divided into two equal independent groups (test set and validation set), and 24 patients with fibroadenoma. All patients were operated on in the Department of Breast Pathology of the National Medical Research Center for Obstetrics, Gynecology, and Perinatology. Breast cancer tissues were obtained from the patients of the study group, and fibroadenoma tissue and healthy breast tissue were obtained from the patients of the control group. All samples were stabilized in RNAlater and stored at −80 °C. DNA was isolated from the samples using QIAamp DNA Mini Kit spin columns (Qiagen, Hilden, Germany) DNA was modified by bisulfite conversion using an EpiJET Bisulfite Conversion Kit (Thermo Scientific, Inc., Waltham, MA, USA). To assess the relative levels of methylation, we used our modified MS–HRM (Methylation Sensitive–High Resolution Melting) method, which has a number of benefits compared to the one proposed by Wojdacz et al. [15]. The modification of the MS–HRM was aimed at creating a protocol the for assessment of methylation of long fragments of CpG islands (500–600 nucleotides) containing a large number of CpG sites. The main difference between the study of methylation in long fragments and an analysis when only one CpG site is studied is the absence of the possibility of using sequences obtained from fully methylated or unmethylated DNA fragments as positive and negative controls (when all CpG sites are represented as CG-dinucleotides or TG-dinucleotides, respectively). Table 6 presents the results of the methylation assessment of the *GCM2* gene by the pyrosequencing of one of the samples. The methylation at different sites ranged from 3% to 62%.

This indicates that most of the molecules in this sample would show partial methylation of CpG sites in various variants. The melting curves from such molecules are not comparable to those obtained from standard samples including only CG or TG dinucleotides (they will cross simultaneously with both standards). Therefore, we had to abandon the control standards and estimate methylation in relative units rather than in percentages. To estimate the methylation level, we chose a single temperature at which the curves obtained from the studied samples diverged from each other as much as possible. The sample with the lowest methylation level was taken as zero. It should be kept in mind that in all samples (both cancer and control), a part of the CpG sites is methylated—as can be seen in Figure 3. As an assay validity control, we performed agarose gel electrophoresis of each sample to ensure that the target product was investigated and that there was no additional influence on the melting curves by a non-specific product or degraded DNA. To show that our results reflected the level of methylation, we determined the percentage of methylation by pyrosequencing each sample (96 samples) for five genes; this confirmed that our method reflects the differences in methylation levels between samples quite accurately. The protocol of our proposed methylation estimation method is presented below. The fragment lengths and the number of CpG sites are presented in Table 7.

A modified MS–HRM protocol.

Amplification of fragments of CpG islands of the studied genes was carried out according to the following protocol: 95 °C—5 min; (95 °C—15 s, 60 °C—30 s, 72 °C—45 s) ×30; (95 °C—15 s, 50 °C—30 s, 72 °C—45 s) ×25. Reducing the annealing temperature from 60 °C to 50 °C after the 30th cycle gave an increased product yield compared to single-step amplification. Primers are shown in Table 7. PCR was performed using the CFX96 system (Bio-Rad, Hercules, CA, USA);Intercalating dye EVAGreen (Syntol, Moscow, Russia) was added to the derived product. Supplementation of EVAGreen in step 1 can lead to PCR inhibition;The melting curve was generated according to the following protocol: 95 °C—30 s; 60 °C—10 min; melting analysis was performed in the temperature interval from 65 °C to 90 °C with increments of 0.2 °C;After the construction of the melting curve, the quality of the PCR product was checked by electrophoresis in a 1.5% agarose gel. Degraded DNA may show false positive results.

Methylation levels were assessed by melting curves using Precision Melt Analysis Software, version 3 (Bio-Rad, Hercules, CA, USA). We did not use standards of different methylation levels when assessing the HRM results, since the standards proposed by Wojdacz et al. [15] provide accurate results for methylation levels only if one CpG site is present in the PCR product. To compare the data, the curves were normalized according to the sample with the lowest level of methylation. Methylation was quantified with relative fluorescence units (RFU) at the temperature of the maximum peaks of the HRM curves (values are shown in charts as relative units). The temperature of the maximum peaks was manually determined for each gene by the researchers based on the appearance of the melting curves (Figure 5).

A comparative analysis was performed according to the following protocol:First, 0.3 μL of the PCR product of the studied gene was taken;NEST-PCR was performed with the primers shown in Table 8. The amplification program was as follows: 95 °C for 5 min: (95 °C for 15 s., 50 °C for 30 s., 72 °C for 45 s.) ×12. NEST-PCR was required to shorten the length of the product to a size suitable for pyrosequencing;A melting curve of the product obtained in step two was generated, and the MS–HRM results were processed according to the protocol mentioned above;Sequencing of the product obtained in step two was performed on a PyroMark Q48 (Qiagen, Hilden, Germany) in accordance with manufacturer’s instructions;The results of the MS–HRM and pyrosequencing were compared using Spearman’s rank correlation coefficient.

Statistical analysis and diagram construction were performed using R (version 4.0.2, R Foundation for Statistical Computing, Vienna, Austria) and OriginPro 8.5 (OriginLab Corporation, Northampton, MA, USA software. The function ci.auc (pROC package) was used to calculate the confidence intervals using the bootstrap method—95%CI, 10,000 bootstrap iterations. The Mann–Whitney U test was used to determine the significance of differences for continuous values. Correlations were assessed using Spearman’s rank correlation coefficient (rs). Data are presented as median (upper quartile; lower quartile). Qualitative characteristics are presented as percentages (number of samples). We applied logistic regression and receiver operating characteristic (ROC) curve analysis separately to assess the diagnostic value of the studied parameters. Differences were considered statistically significant at *p* < 0.05.

## 5. Conclusions

In this work, we demonstrated that the proposed simple, modified MS–HRM method allows the validation of results for DNA methylation in breast cancer in large sets of samples. It was found that the level of methylation depended on the size, subtype, and histological characteristics of the tumor; this dependence should be considered when determining study samples for the development of diagnostic test systems. In our study, we used a set of 96 tumor samples of less than 2 cm; in this set, the assessment of the methylation of the *GCM2*, *ITPRIPL1*, *CACNA1E*, and *DLGAP2* genes most accurately distinguished cancer from healthy tissues.

## 6. Patents

Method for the diagnosis of the methylation of small breast cancer tumors is protected by Russian Patent no. RU 2789200 C1: Method for diagnosing breast cancer by the level of methylation of the *ZNF177* and *CCDC181* genes in small tumors [16].

## Figures and Tables

**Figure 1 ijms-24-12660-f001:**
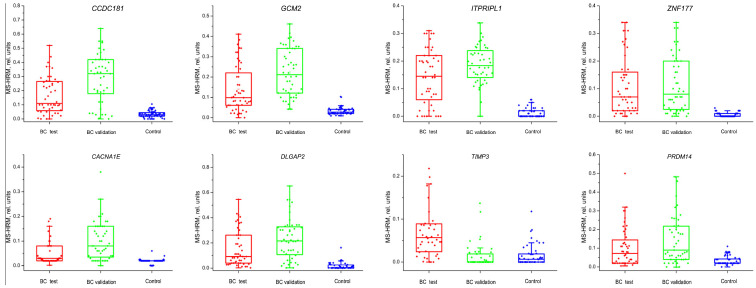
The levels of methylation of the studied genes in breast cancer (test set and validation set) compared to control samples (healthy breast tissue, fibroadenoma).

**Figure 2 ijms-24-12660-f002:**
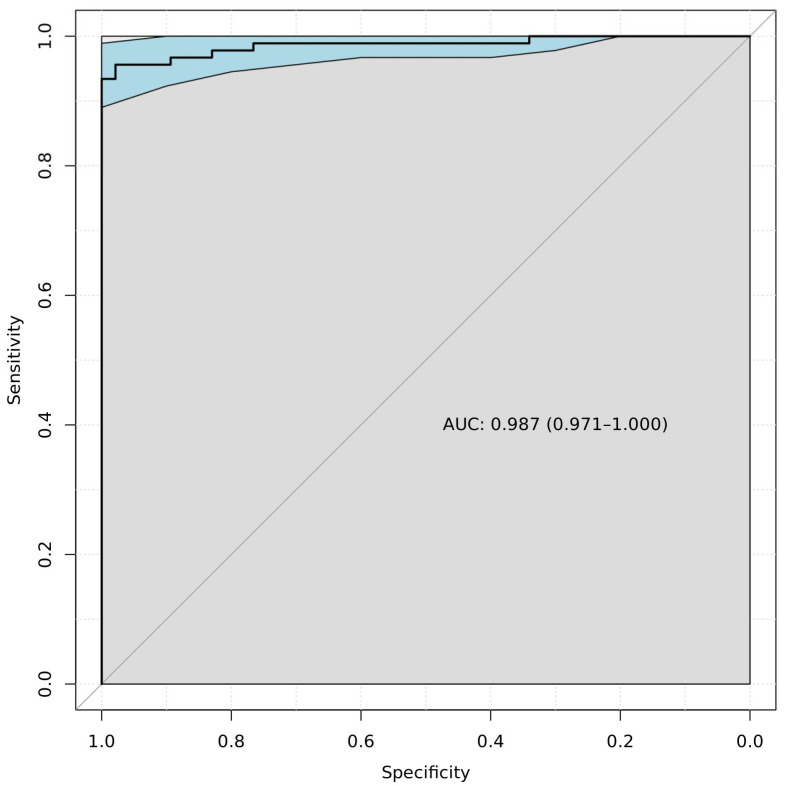
ROC curve for the diagnostic model which included the *GCM2*, *ITPRIPL1*, *CACNA1E*, and *DLGAP2* genes. The blue band corresponds to the 95% CIs, which was calculated using the bootstrap method. Gray color indicates the range of sensitivity and specificity values.

**Figure 3 ijms-24-12660-f003:**
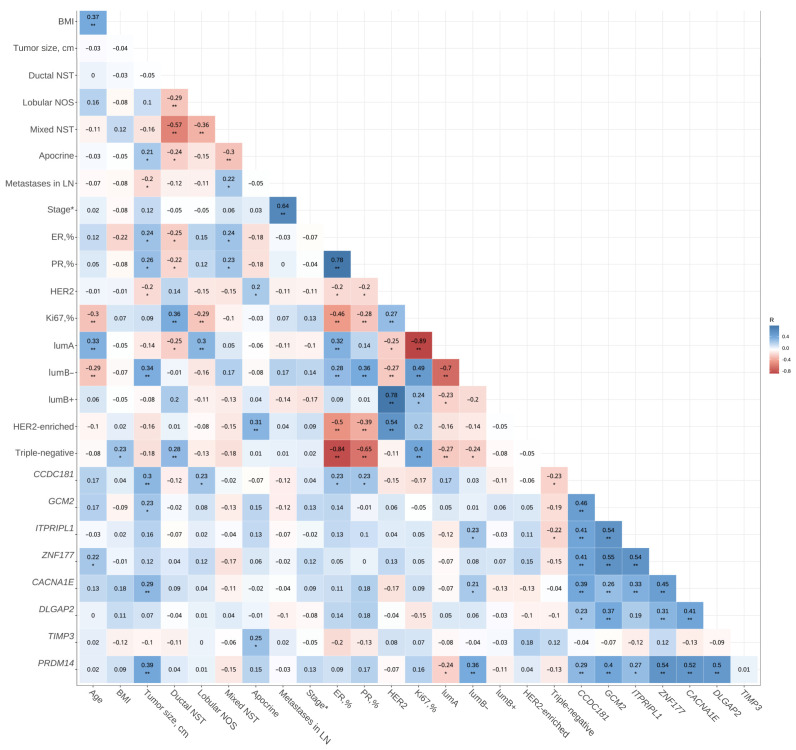
Correlation between tumor characteristics and methylation status, *—*p* < 0.05, **—*p* < 0.01. * Stage, formalization: Stage I: 1, Stage IIa: 2, Stage IIb: 3, and Stage IIIa: 4.

**Figure 4 ijms-24-12660-f004:**
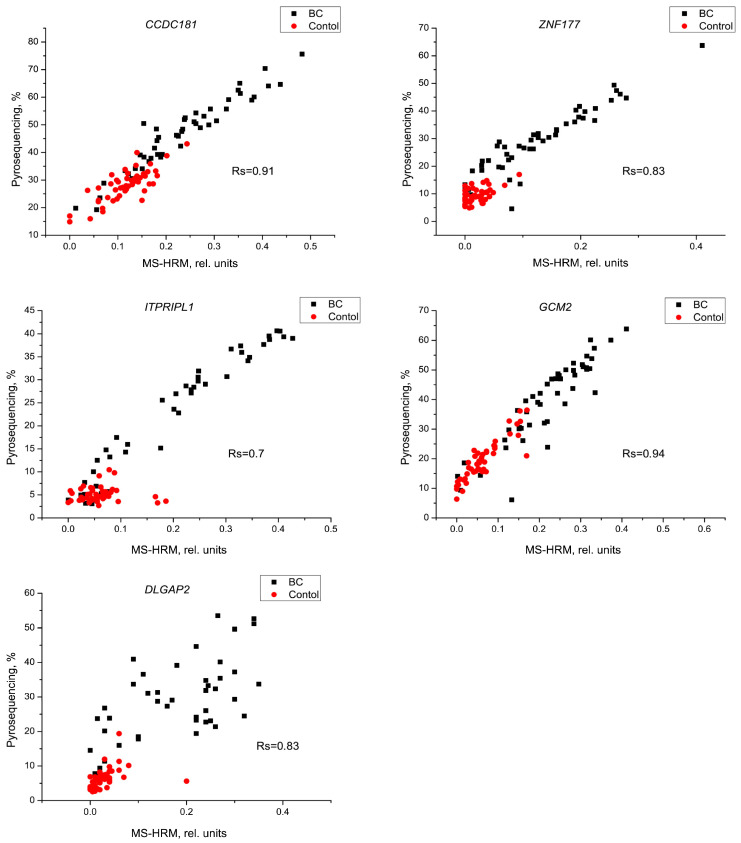
Comparison of methylation levels of the *CCDC181*, *ZNF177*, *DLGAP2*, *GCM2*, and *ITPRIPL1* genes using modified MS–HRM and pyrosequencing.

**Figure 5 ijms-24-12660-f005:**
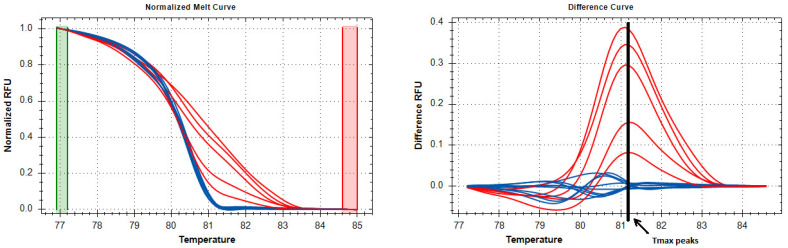
Temperature peaks at which the RFU values were compared. Blue lines are melting curves for unmethylated samples. Red lines are melting curves for samples with different levels of methylation.

**Table 1 ijms-24-12660-t001:** Age and BMI of the studied groups.

Parameters	Control	Test Set	*p*-Value	Validation Set	*p*-Value
Age	47 (42; 52)	51 (41; 60)	0.22	50 (45; 57)	0.11
BMI	26 (23; 28)	28 (25; 32)	0.06	28 (23; 33)	0.07

*p*-values are shown when comparing groups with controls.

**Table 2 ijms-24-12660-t002:** Tumor subtypes and histological characteristics of breast cancer tumors.

Morphological Forms	Test Set, (*n* = 48)	Validation Set, (*n* = 48)
Invasive ductal breast carcinoma of no special type	25 (12)	37.5 (18)
Invasive lobular carcinoma not otherwise specified	16.7 (8)	14.6 (7)
Invasive mixed breast cancer of no special type	45.8 (22)	37.5 (18)
Invasive apocrine breast cancer	12.5 (6)	10.4 (5)
Subtype		
Luminal A (LumA)	33.3 (16)	54.2 (26)
Luminal B (LumB−)	37.5 (18)	37.5 (18)
Luminal B-like (LumB+)	12.5 (6)	2.1 (1)
HER2-enriched	6.2 (3)	0
Triple-negative	10.4 (5)	6.2 (3)
Tumor length, cm	1.5 (1.3; 1.8)	1.7 (1.4; 1.9)
Estrogen receptor (ER) positive	83.3 (40)	93.7 (45)
Progesterone receptor (PR) positive	84.4 (38)	91.1 (41)
Ki67, %	23 (14; 47)	17 (12; 46)
stage, formalization *	1 (1; 2)	1 (1;2)
metastases in lymph node	37.5 (18)	10.4 (5)

* Stage, formalization: Stage I: 1, Stage IIa: 2, Stage IIb: 3, and Stage IIIa: 4. Data are presented as median (upper quartile; lower quartile). Qualitative characteristics are presented as percentages (number of samples).

**Table 3 ijms-24-12660-t003:** The levels of methylation of the studied genes in breast cancer (test set).

Gene	Methylation Breast Cancer Control		*p*-Value
*POM121L2*	0 (0; 0.04)	0 (0; 0.01)	0.23
*KCNQ1*	0.02 (0; 0.19)	0.03 (0; 0.14)	0.97
*MAST1*	0.13 (0.04; 0.17)	0.14 (0.07; 0.19)	0.3

**Table 4 ijms-24-12660-t004:** Diagnostic value of genes according to ROC analysis.

Gene	AUC (95% CI)	
	Test Set	Validation Set
*CCDC181*	0.85 (0.77–0.93)	0.89 (0.82–0.97)
*GCM2*	0.86 (0.77–0.94)	0.98 (0.96–1)
*ITPRIPL1*	0.88 (0.803–0.95)	0.98 (0.95–1)
*ZNF177*	0.91 (0.86–0.97)	0.93 (0.87–0.98)
*CACNA1E*	0.77 (0.69–0.86)	0.88 (0.81–0.95)
*DLGAP2*	0.91 (0.85–0.96)	0.95 (0.91–0.99)
*TIMP3*	0.84 (0.757–0.92)	0.57 (0.47–0.68)
*PRDM14*	0.72 (0.62–0.82)	0.79 (0.70–0.89)

**Table 5 ijms-24-12660-t005:** Comparison of diagnostic value of the studied genes methylation by pyrosequencing and MS–HRM according to the ROC analysis.

Gene	AUC (95% CI)	
	Pyrosequencing	MS–HRM
*GCM2*	0.88 (0.8–0.95)	0.88 (0.8–0.95)
*DLGAP2*	0.96 (0.93–0.995)	0.84 (0.75–0.92)
*ITRIPL*	0.85 (0.77–0.94)	0.78 (0.68–0.88)
*CCDC181*	0.9 (0.84–0.97)	0.84 (0.76–0.92)
*ZNF177*	0.98 (0.95–1)	0.89 (0.83–0.96)

**Table 6 ijms-24-12660-t006:** Methylation of CpG sites of the GCM2 gene assessed by the pyrosequencing method.

CpG Site	1	2	3	4	5	6	7	8	9	10	11	12
Methylation (%)	62	40	28	37	33	32	9	17	37	4	3	10

First CpG site located after 233 bp from transcription start site (TSS).

**Table 7 ijms-24-12660-t007:** Primers for MS–HRM.

Gene	Annealing Temperature	Amplicon Length	Number of CpG Sites	Forward Primer	Reverse Primer
*CCDC181*	60	432	26	GGAGTGAGGTGTTTTGGGGTTTA	CTAATATATAAATTTCCTTCATCTTAT
*GCM2*	60	528	36	GTTTTTGGGATTGTGGTGGAG	AATACCATTCTCCCTCCTTC
*ITRIPL1*	63.5	210	13	GGATAAGTATGGTTTATAATTTAGG	CCAAAAACCCATCCTACCTCGAAATATCTCC
*POM121L2*	58	828	67	GGAAATTTTTAATTAGTTGTT	CCTCCTCACAAATCTATACC
*KCNQ1*	60	494	51	GTAGGAGTAAGTAGGGGAGATGTAGA	AACAACCACTACTACCAAC
*MAST1*	63	605	75	TTTATGGGGGTATTAGGAGGT	ACCCCAACCCCATCCCCCTA
*PRDM14*	58	654	41	TTATTTAGTTAAGAGGAAGTAG	ACCTTCTAAAACAAACAATATTAC
*DNM2*	61	401	47	GGTTAGAGTTGTTATTTGGATTTGT	ACCAACCAAATAACAAACTTCACC
*ZNF177*	63	345	23	GGGTAGTTTATTTTTTTTAGTTGTTGGT	CACATAAACCCACTTACCTCCTC
*AP2M1*	59	691	63	GTGTATGTTGGAATGTGTAT	ACACAAAAAAATATCACTATCCTAC
*CPEB4*	57	226	23	GGTTTTGAGGAGAAGGATTTAGT	AACTTTTATTTCTCCTCA
*DLGAP2*	57	199	34	GGAGGTTTTGTTTTTAGTATTTAAG	AACAAAATCAACCTTTCTAAAC
*CACNA1E*	62	380	29	TAGAGTTGGAGTTTAGGAAGGGGTTAT	CATACATACCCACCACCC
*SSH1*	62	341	40	AGTTTTTTAAAGTGTTGGGATTATAGG	CACTACATACACAATCCCTACAA
*TIMP3*	60	165	15	GGTTAGAGATATTTAGTGGTTTAGGTGG	TTCAAATCCTTATAAAAAATAATACC

**Table 8 ijms-24-12660-t008:** Primers for pyrosequencing.

Gene	Annealing Temperature	Amplicon Length	Number of CpG Sites	Forward Primer	Revers Primer
*GCM2*	50	99	12	TTTATTTTTTAGAATTTTGG	CCAACTAAACTACATCC-biotin
*ITPRIPL1*	50	122	11	GGAGTAGGGTTATT	TCAATATTAATAAAAACAC-biotin
*DLGAP2*	50	97	15	GAGTTTTTTTGGG	CAAAATCAACCTTTCTAA-biotin
*CCDC181*	50	107	6	TTTTGGGGTTTAATTTGTG	ACCTACTTCCAATCTTCAAC-biotin
*ZNF177*	50	100	7	GGGTAGTTTATTTTTTTTAGTTGTTGGT	AACAACCCTTTCTCAACTACA-biotin

## Data Availability

The data that support the findings of this study are available from the corresponding author, upon request.

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
