# Peer review of "Methylation Profile of Small Breast Cancer Tumors Evaluated by Modified MS–HRM"

_ijms, 2023, doi:10.3390/ijms241612660_

Round 1

Reviewer 1 Report (Previous Reviewer 1)

Please test an additional 100 breast cancers vs fibroadenomas to confirm the high AUC using the same methylation panel.

Author Response

Dear Reviewer, Thank you very much for your high estimate of our work.  You recommend to increase the sample in order to confirm the AUC we have obtained. Indeed, correlation analysis showed that the methylation level depends on tumor characteristics such as size and subtype, so we cannot exclude that the accuracy of the analysis may change with a different sample. Unfortunately, we will not be able to find another 100 samples of small tumors in a short time. When we first submitted the paper to IJMS, the cancer group consisted of 50 samples and we were advised to increase the number by 2 times. It took us almost half a year to collect 50 more samples. For the reported study we decided to use statistical methods to confirm the stability of AUС. We used a bootstrap method to calculate CI for ROC curves (Carpenter and Bithell 2000;Efron 1987). This is a simple but powerful method for estimating confidence intervals without the need to repeat the experiment (Xia et al., 2013). This is achieved by designing the groups of different samples (called "replicated samples") by randomly sampling and replacing the original data set. This technique allowed us to avoid dependence of the result on the characteristics of the original sample. Using bootstrap method, we obtained CI= 0.9675-0.9988, which confirms the stability of the AUC.

Reviewer 2 Report (New Reviewer)

Authors have selected their target genes which are differentially methylated in small breast cancer from previously published papers and applied their modified MS-HRM method to validate those results. They showed that  effectiveness of their method to differentiate the methylation profile for small breast cancer.

There is positive and negative controls missing in their analysis which should ascertain that the outcomes of analysis are reliable.

There are typographical errors in the manuscripts that should be corrected carefully.

Quality of language is acceptable, but the typographical errors should be corrected.

Author Response

Dear Reviewer, Thank you very much for your high estimate of our work.  Your main remark is the lack of positive and negative controls. Unfortunately, creating such controls when an MS-HRM analysis considers an extended fragment with a large number of CpG sites is a difficult task. When there is only one CpG site between the forward and reverse primers, it is possible to synthesize two products as a control. The negative control containing a TG dinucleotide and the positive control containing a CG dinucleotide can be synthesized. Which is applied in the classic MS-HRM variant. By mixing these controls in different proportions we can obtain melting curves corresponding to different concentrations of the methylated part of DNA. However, in case we study two or more CpG sites, e.g. CGCG, it turns out that a significant part of DNA molecules has methylation in only one position, i.e. the sequence in the PCR product will be CGTG or TGCG. We found this when we sequenced the PCR products using NGS.   The melting curves of such products are not comparable to those obtained from the CGCG and TGTG standards (they will cross simultaneously with both standards). Therefore, we had to abandon the reference standards and estimate methylation in relative units instead of percentages. To estimate the methylation level, we chose the only temperature at which the curves obtained from the tested samples differed as much as possible from each other.  The sample with the lowest methylation level was taken as zero. Note that in all samples (both cancer and control) part of the CpG sites are methylated. This is shown in Figure 4, where ms-hrm and pyrosequencing are compared. To ensure that this is the target product and there is no additional influence on the melting curves of nonspecific product or degraded DNA, we performed agarose gel electrophoresis for each sample. To show whether our results reflect the level of methylation, we determined the percentage of methylation by pyrosequencing in each sample for 5 genes. And this confirmed that our method accurately reflects the differences in methylation level between samples.

Obviously, this aspect was not sufficiently reflected in our publication and, therefore, we expanded the methodological section in the article.     

This manuscript is a resubmission of an earlier submission. The following is a list of the peer review reports and author responses from that submission.

Round 1

Reviewer 1 Report

In the title, the term breast cancer tumors is confusing. Since you analyzed both benign and malignant tumors, suggest removing "cancer" and just indicate breast tumors. Also in the title, you indicate "small" tumors. This is not as important as N and M stage. If all tumors are N0 and M0, please indicate this, and call all of the breast cancers stage 1. If they are not, please indicate TNM for each breast cancer. What was the subtype (ductal, lobular, metaplastic) for each breast cancer? Did any patient receive neoadjuvant chemotherapy? Clinical information is inadequate.

Your AUC results are quite impressive, but sample size inadequate to draw conclusions. Please repeat your studies with a fresh set of breast cancers and benign tumors of a size at least equal to the currently analyzed cohort, to see if AUC holds up.

Reviewer 2 Report

The manuscript "Methylation profile of small breast cancer tumors" by Krasnyi et. al. describes promoter methylation in selected genes of small size breast cancers. Authors indicate that markers CCDC181 and ZNF177 together have a diagnostic value for this tumor subgroup.

Conclusions are based on a small set (#48) of breast cancers, obtained from different datasets. It is difficult to come to a conclusion from such a small set and varied selection of cancer samples. Overall data is not convincing for the following reasons.

1. Studies are based solely on promoter methylation and the title indicates methylation, i. e. could include intragenic methylation which is not included.

2. NCBI Blast N nucleotide search analysis of forward and reverse primer sequences of CCDC181, PRDM14, ZNF177, and TIMP3 (only markers checked using the website), do not match to these genes. It is possible the NIH website does not cover promoter regions and thus there are no matches to the genes. Considering this possibility, authors should have included sequence information of PCR products to check for gene authenticity.

3. Although mentioned, analysis of PCR products in agarose gels should have been included to correspond to the product size shown in the Table. In the absence of the gel and sequence data, it is not possible to evaluate the significance of the reported genes in breast cancer.

4.  Sanger database of Breast cancer indicates that there is methylation of CCDC181 in 158/707 (22.35%). A comparison could be made to this information to show whether there is higher promoter methylation for CCDC181 and other genes in the analyzed data set.

5. Although it is expected that there will be suppression of transcription with promoter methylation of tumor suppressor genes, there is no data provided comparing the percentage of CpG promoter methylation and transcription of the genes selected as diagnostic markers. Intragenic methylations could provide supportive information for the association of promoter methylation and transcription suppression.

Reviewer 3 Report

Aim is not that clear

The title is concise not informative

need bioinformatics background and basis for the work

Limitations, strengths and future prospectives and recommendations are missing.

Confirmation via reactome and string are missing

Study design to be mentioned

Rational for DNa methylation choice

More recent references are needed

Reviewer 4 Report

This study analyzed small malignant breast tumors and compared the methylation pattern of 15 genes with healthy tissue and fibroadenoma. The methylation patterns were profiled using their modified MS-HRM method, and the results were validated using pyrosequencing. Significant differences were found for eight genes, and two genes were identified for the most accurate diagnosis. This is a nice short paper.

1.       The criteria for selecting the 15 genes needs to be clarified more.

2.       Fig.1 also should show the results of the rest of the seven genes that did not show significant differences between tumor and control.  

3.       Figures 1, 2, and 3: What is the unit of the “MS-HRM, rel. unit”? What are the two numbers along the axis?

4.       The “Conclusions” section should be before the “Materials and Methods” section, or the “Materials and Methods” section should be before the “Results” section.